# Duchenne Muscular Dystrophy Patient iPSCs—Derived Skeletal Muscle Organoids Exhibit a Developmental Delay in Myogenic Progenitor Maturation

**DOI:** 10.3390/cells14131033

**Published:** 2025-07-07

**Authors:** Urs Kindler, Lampros Mavrommatis, Franziska Käppler, Dalya Gebrehiwet Hiluf, Stefanie Heilmann-Heimbach, Katrin Marcus, Thomas Günther Pomorski, Matthias Vorgerd, Beate Brand-Saberi, Holm Zaehres

**Affiliations:** 1Department of Anatomy and Molecular Embryology, Institute of Anatomy, Faculty of Medicine, Ruhr University Bochum, 44801 Bochum, NRW, Germany; urs.kindler@rub.de (U.K.); lampros.mavrommatis@rub.de (L.M.); franziska.kaeppler@rub.de (F.K.); dalya.hiluf@rub.de (D.G.H.); beate.brand-saberi@rub.de (B.B.-S.); 2Department of Molecular Biochemistry, Faculty of Chemistry and Biochemistry, Ruhr University Bochum, 44801 Bochum, NRW, Germany; thomas.guenther-pomorski@rub.de; 3Department of Neurology with Heimer Institute for Muscle Research, Faculty of Medicine, University Hospital Bergmannsheil, Ruhr University Bochum, 44789 Bochum, NRW, Germany; matthias.vorgerd@bergmannsheil.de; 4Institute of Human Genetics, NGS Core Facility, Faculty of Medicine, University of Bonn, 53127 Bonn, NRW, Germany; sheilman@uni-bonn.de; 5Medical Proteom Center, Faculty of Medicine, Ruhr University Bochum, 44801 Bochum, NRW, Germany; katrin.marcus@rub.de

**Keywords:** skeletal muscle, organoids, myogenesis, myogenic progenitors, satellite cells, FAPs, Duchenne muscular dystrophy, human-induced pluripotent stem cells, scRNA-seq

## Abstract

Background: Duchenne muscular dystrophy (DMD), which affects 1 in 3500 to 5000 newborn boys worldwide, is characterized by progressive skeletal muscle weakness and degeneration. The reduced muscle regeneration capacity presented by patients is associated with increased fibrosis. Satellite cells (SCs) are skeletal muscle stem cells that play an important role in adult muscle maintenance and regeneration. The absence or mutation of dystrophin in DMD is hypothesized to impair SC asymmetric division, leading to cell cycle arrest. Methods: To overcome the limited availability of biopsies from DMD patients, we used our 3D skeletal muscle organoid (SMO) system, which delivers a stable population of myogenic progenitors (MPs) in dormant, activated, and committed stages, to perform SMO cultures using three DMD patient-derived iPSC lines. Results: The results of scRNA-seq analysis of three DMD SMO cultures versus two healthy, non-isogenic, SMO cultures indicate reduced MP populations with constant activation and differentiation, trending toward embryonic and immature myotubes. Mapping our data onto the human myogenic reference atlas, together with primary SC scRNA-seq data, indicated a more immature developmental stage of DMD organoid-derived MPs. DMD fibro-adipogenic progenitors (FAPs) appear to be activated in SMOs. Conclusions: Our organoid system provides a promising model for studying muscular dystrophies in vitro, especially in the case of early developmental onset, and a methodology for overcoming the bottleneck of limited patient material for skeletal muscle disease modeling.

## 1. Introduction

Duchenne muscular dystrophy (DMD) and Becker muscular dystrophy (BMD) are the most common muscle diseases out of more than 40 genetic skeletal muscle dystrophies [1,2,3]. DMD was first described by Sir Charles Bell in 1830, Giovanni Semmola in 1834, and Gaetano Conte in 1836, who reported a progressive weakness of muscles in boys and muscular hypertrophy. Later, Guillaume Benjamin Amand Duchenne described the first detailed DMD case in 1861 and gave this specific form of muscular dystrophy its name [4,5]. DMD is transmitted on the X chromosome and consequently found with full transmission only in male children, while females carrying the DMD mutation are mostly asymptomatic [6]. Worldwide, 19.8 out of every 100,000 males born alive are affected by DMD, which manifests in multiple organs, not just skeletal muscle, and naturally affects many phases of their lives [7,8]. Due to the lack of dystrophin in DMD patients, the muscle fibers are very prone to damage and degeneration. The consequence is a cycle of increased inflammation, fibrosis, and regeneration [9].

Satellite cells (SCs) are adult skeletal muscle stem cells located between the basal lamina and the muscle fiber membrane. They are essential for adult skeletal muscle tissue maintenance and the regeneration and growth of muscles. SCs can proliferate, differentiate, and fuse into myofibers that regenerate injured muscle after their activation [10].

SCs are resting stem cells expressing a specific set of markers and transcription factors, such as PAX7, MYF5, or SPROUTY1. PAX7 is the most prominent factor, and it is usually used as an identification marker. When SCs are in the quiescent state, metabolic and mTOR activity is decreased [11,12]. Upon injury, SCs enter the cell cycle and start expressing MYF5 as well as cell cycling genes such as KI67 and TOP2A. Here, symmetric expansion, asymmetric cell division, and commitment can be distinguished [12]. It has been reported that the balance between pool-reserved programs like asymmetric division is regulated by interleukin-6 or p16 (INK4a) [12]. After the activation of SCs, Pax7 is downregulated, and the cells start to express MRFs such as MYF5 and MYOD and begin to enter the differentiation program [13,14].

The regeneration of myofibers takes place through the fusion of individual satellite cells with muscle fibers to form a multinucleated syncytial structure. Gene products can be freely transported within each muscle fiber, making satellite cells good carriers for corrected gene products in muscular dystrophies [15].

One study demonstrated that satellite cells in DMD are affected by constant activation, which is a reasonable conclusion considering the increased quantity of committed progenitors in DMD mice [16]. Another study showed that there is constant progression of muscle loss and an elevated number of SCs in DMD patients [17]. This study examines whether the two discoveries are related to or underlie different aspects of DMD.

As recently shown, dystrophin is highly expressed in activated SCs and is polarized when they undergo cell division [18]. In this context, the Par complex, consisting of cell polarity-regulating kinase (MARK2), PARD3, PAR6, and atypical protein kinase C (aPKC), plays an important role. During asymmetric division, MARK2 binds to dystrophin at the eighth and ninth spectrin-like repeats [19] and moves to the basal pole in the SCs, while the Par complex polarizes at the apical pole [18]. Thus, the Par complex is integrated at the apical side and phosphorylates Numb. Numb is a Notch inhibitor that moves to the basal region. Here, the apical daughter cell develops into a differentiated cell, whereas the basal daughter cell stays in the self-renewal state [20,21]. Investigations using a DMD mouse model (mdx mouse) showed that Mark2 expression is reduced and that polarization is lost because Pard3 is equally distributed, resulting in impaired muscle regeneration potential because the mitotic spindle symmetrically dissects the Pard3-containing region [18].

Over the course of several years, various protocols have been proven effective in the directed differentiation of pluripotent cells into skeletal muscles and have been used to study the developmental stages of muscle differentiation [22,23,24,25,26,27]. Importantly, iPSCs need to be guided toward the paraxial mesoderm, a process that can be achieved via WNT pathway activation and the inhibition of the BMP and FGF pathways [22,23,28,29]. WNT activation supports the specification of the early paraxial mesoderm, and BMP inhibition prevents a shift in the fate of the paraxial mesoderm toward the lateral-plate mesoderm. The transition from the posterior to the anterior somitic mesoderm can be confirmed via PAX3 expression [23]. As stated by the protocol developed by Chal and colleagues, treatment with trophic factors such as bFGF, IGF, and HGF allows the PAX3 population to induce the myogenic program [24]. One major aspect of this protocol is the fact that the PAX7 population decreases over time during differentiation. One study revealed that the population decreased by 10% each purification cycle after FACS was used to sort PAX7-positive cells after two weeks of culture [30]. This finding indicates that the cultivation of myogenic progenitors (MPs) over extended passages becomes difficult in 2D differentiation.

Recently, 3D differentiation has gained traction in research for its ability to mimic the environmental heterogeneity and interactions among cell types and within tissues. In general, the microenvironment in which stem cells are regenerated through signals in their surroundings, thereby helping to maintain and regenerate tissue, is called the stem cell niche. In the case of SCs, various components are required to establish the stem cell niche, such as ECM proteins but also cells like fibro-adipogenic cells, myofiber endothelial cells, pericytes, and macrophages, which influence SC fate through signals [31,32]. Besides the establishment of the stem cell niche, the processes and conditions of these organoids often resemble those of early embryonic development as well as the fetal maturation of cells, processes essential for the first pathological signs and the early phenotype in disorders like DMD [33,34]. Recent protocols have also achieved 3D muscle differentiation [35,36,37,38,39,40,41]. While some authors have used a pre-differentiation step to generate myogenic progenitors, endothelial cells, neural progenitor cells, and pericyte-like cells that are simultaneously embedded to ensure cell diversity within the 3D construct [36,39], other protocols started from iPSCs that are either clustered in a ball-like structure using hanging-drop technology or ultra-low attachment plates. WNT activation, FGF treatment, and BMP inhibition are also necessary to induce the activity of the paraxial mesoderm. In the 3D skeletal muscle organoid (SMO) system established in our lab [37,38], the induction of presomitic mesoderm maturation was achieved via retinoic acid treatment. Shortly after treatment with SHH, dermomyotome was induced [42]. Subsequent treatment with the trophic factors IGF and HGF favored myogenic differentiation while avoiding neural fate cell differentiation. Shifting top HGF as a single growth factor triggers constant activation of the proliferation and division of myogenic progenitors [12].

As we addressed the specific maturation and characterization of myogenic progenitors in their dormant, activated, and committed states—in which they resemble SCs—within an SMO system in our previous study [38], we provide and evaluate herein the scRNA-seq datasets of three DMD patient iPSC-derived SMOs, with a focus on alterations in myogenic progenitor cell populations.

## 2. Materials and Methods

### 2.1. Human-Induced Pluripotent Stem Cell (iPSC) Lines

Duchenne muscle dystrophy patient iPSC lines:“DMD1” (DMD_iPS1) (passage 30–50) [43]“DMD2” (DMD_iPS2) (passage 30–50) [43]“DMD3” (iPSCORE_65_1, UCSD061i-65-1, WiCell, #WB60393) (passages 20–50) [44]Wild Type iPSC line:“WT” Cord Blood iPSC (CB CD34+ iPSC) (passages 23–50) [45]

### 2.2. hiPSC Culture

Human-induced pluripotent stem cells (hiPSCs) were maintained in TeSR™-E8™ (StemCell Technologies, 05990, Vancouver, BC, Canada) on 4% Matrigel GFR (Corning, 354230, Corning, NY, USA) coated dishes or on 1% Geltrex™ Reduced Growth Factor Basement Membrane Matrix (Gibco™, #A1569601, Waltham, MA, USA) coated dishes at 37 °C and 5% CO_2_. The media were exchanged daily.

At a confluency of 80%, iPSCs were passed in a ratio of 1/4 to 1/6 and plated on newly coated culture dishes. During the split, the cells were incubated with TrypLE™ Select (Gibco™, Thermo Fisher Scientific, Waltham, MA, USA, 12563011) in a humidified environment for 3–5 min at 37 °C with 5% CO_2,_ followed by mechanical dissociation by pipetting. Dissociation was stopped with DMEM/F12 media (Gibco™, 21331020, Waltham, MA, USA), and the cells were pelleted at 400 g for 5 min at room temperature. The cells were replated in TESR-E8 supplemented with 10 μM Rock inhibitor (Y-27632, StemCell Technologies, 72304, Vancouver, BC, Canada), suppressing apoptosis. The Rock inhibitor was removed 24 h after the media were replaced.

Both Matrigel (Corning, New York, NY, USA) and Geltrex (Thermo Fisher, Waltham, MA, USA) were utilized for hiPSC culture, depending on the specific requirements of individual cell lines. For direct comparisons (e.g., WT vs. DMD), paired lines were cultured on the same matrix batch.

### 2.3. Skeletal Muscle Organoid Differentiation

Three-dimensional skeletal muscle organoid differentiation was performed as described in detail in [38,46].

Briefly, human iPSCs were dissociated at 75% confluency using TrypLE and resuspended in TESR-E8 supplemented with 10 µM of the ROCK inhibitor. Embryoid bodies (EBs) were formed via the hanging-drop method (4000 cells/20 µL) containing 4 mg/mL polyvinyl alcohol (PVA) to enhance viscosity. After 24 h, the EBs were embedded in Matrigel^®^ (30 µL/EB) and cultured for 12 weeks in differentiation media. Media changes were performed every 2–3 days.

The basal media consisted of DMEM/F12 basal medium (Gibco) supplemented with 1× glutamine (Gibco), 1× non-essential amino acids (Gibco), and 100× ITS-G (Gibco). After day 14, the basal media consisted of DMEM/F12 basal medium (Gibco) supplemented with 1× glutamine (Gibco), 1× non-essential amino acids (Gibco), and 100× ITS-X (Gibco).

The basal media were freshly supplemented with mixtures of growth factors at various concentrations and in different orders, as outlined in Appendix A.

Documentation over the time of differentiation was performed by brightfield images (Appendix A). Morphological examinations were performed on an OLYMPUS CKX41 microscope with a 4× objective and phase contrast. Images were processed using ImageJ version 1.54f (Wayne Rasband/NIH, Bethesda, MD, USA). Statistical analyses were conducted using GraphPad Prism version 9 (GraphPad Software, Inc., Boston, MA, USA) software.

### 2.4. scRNA-seq Sample Preparation and Sequencing 

For sample preparation, two 12-week-old organoids each from WT, DMD1, DMD2 and DMD3 were dissociated with a cocktail of 36 mg of papain, 8 mg of EDTA, and 8 mg of L-Cystein in 20 mL of DMEM/F12. After 1 h of dissociation at 37 °C and under normoxic conditions, the organoids were additionally dissociated mechanically by pipetting once after the incubation time. After dissociation was stopped using DMEM/F12, the single-cell suspension was filtered through a 40 μm cell strainer to remove large cells and non-dissociated cell clumps. The cells from the same cell line were pooled. The range of cells used was from 6.6 × 10^−5^ to 1 × 10^−6^ cells. The cells were resuspended in 0.04% BSA in 1xPBS. The live/dead ratio of a single-cell suspension was determined by Trypan Blue solution in a Neubauer chamber and a cell counter from Thermo Fisher. Further library preparation and sequencing were performed by Life & Brain GmbH (Bonn, Germany). The cells were processed with the Chromium Next GEM Single Cell 3′ Kit v3.1 (10× Genomics, 1000269, 10x Genomics B.V., Leiden, The Netherlands) and the Chromium Next GEM Chip G Single Cell Kit (10× Genomics, 1000127, 10x Genomics B.V., Leiden, The Netherlands) according to the manufacturer’s instructions. The cDNA library was run on NovaSeq 6000. The variable cell input numbers (6.6 × 10^5^ to 1 × 10^6^) reflected the optimization for maximum cell recovery per organoid sample. Following chip loading, GEM generation occurred, and all GEMs underwent cDNA synthesis. For the 3′ Gene Expression Dual Index Library Construction, 10 μL (25%) of the synthesized cDNA was used. The number of PCR amplification cycles was adjusted based on measured cDNA input to ensure standardized library quality across samples.

### 2.5. Data Analysis Pipeline

#### 2.5.1. Raw Data Processing with Cell Ranger

To align the raw data to the human genome, Cell Ranger version 5 software was used. Default settings were used to make the samples comparable to existing datasets from the organoid system in our lab. The suggested analysis pipeline from 10× Genomics was followed by the provided hg38 (human genome and transcriptome information) to align the reads information with the human transcriptome. The outcome is stored in barcode, matrix, and feature files, which can be imported and processed by bioinformatic packages.

#### 2.5.2. Seurat Data Analysis

To analyze organoid SC RNA-seq datasets, “Seurat” version 4.3 was used [47,48]. The subset of organoid datasets was filtered for cells with more than 250 to 500 and fewer than approximately 1500 to 7500 detected genes, and cells with more than 5 to 10 percent mitochondrial transcript proportions were excluded. The normalizeData function was adjusted using the normalization.method = “CLR” and the margin = 2 command, which normalizes the genes within each cell, not with genes along the cell population. Between 1500 and 2000 highly variable genes (HVGs) were used after confirming this for each organoid dataset.

The sequencing depth, proportions of mitochondrial transcripts, cell cycle effects [49], and genes associated with stress during dissociation [50] were regressed.

External datasets followed the same analysis steps. Default settings, if not stated otherwise in the paper, were applied [30,51,52]. The adult satellite cell datasets were processed by the provided R code according to prior analysis [53]. For the external datasets, the “S.Score”, “G2M.Score”, “Stress”, and “total Count” were regressed out.

#### 2.5.3. Quality Control for Organoid-Derived Datasets

The features were plotted against the cell counts for quality control analysis (Appendix A). This plot helped to identify the lowest and highest borders for features within each cell. Additionally, the median and MAD (median absolute deviation) were calculated to determine the upper boundaries for the feature cutoffs. Hereby, it is recommended to stay within the range of 3–5 MAD from the median to avoid data points containing more than one cell [54]. The organoid WT dataset GSE147512 was used as a control. Here, the calculated value (6.418–8939) was compared with the literature feature cutoff of 6000 [38]. This confirmed the visual plots that were used to select the features and mitochondrial cutoff (Appendix A).

Cells with >5–10% mitochondrial transcripts were excluded, as elevated mitochondrial RNA serves as a marker of cellular stress or apoptosis [55].

Since there is no common rule to estimate the exact number of HVGs, Seuratobject was converted into SingleCellExperimentobject [56]. In connection with the “scran” package [57], the *p*-value of each cell can be calculated to test the null hypothesis (its variance equaling the trend). All genes with a *p*-value below 0.05 were determined as HVGs and considered significant features (http://bioconductor.org/books/3.12/OSCA/feature-selection.html (accessed on 25 June 2025)). This helps to evaluate the quality of the dataset and gives a general trend for the diversity of these datasets.

The percentage of variation between two continuous principal components (PCs) was calculated to give a more accurate depiction of how many PCs need to be included in the downstream analysis: “RunUMAP”, “FindNeighbors”, and “FindClusters”. Therefore, the continuous PCs that have a change of less than 0.1% are visualized and selected.

#### 2.5.4. Integrative Seurat Analysis

Integrative Seurat analyses were performed to compare (organoid) datasets with one another [58]. Default parameters were applied except for the preprocessing of data before integration. Specifically, preprocessing occurs before the integration of the data, where each dataset is individually trimmed with min_nFeature_RNA between 250 and 500 and max_nFeature_RNA based on 3–5× MAD and normalized using “NormalizeData” with CLR normalization (normalization.method = “CLR”, margin = 2). The datasets were merged, and before the integration, the PC value was set to 30 because no calculation was possible at this step. The “FindVariableFeatures” function was applied with a range of 1500 and 2000 features to each dataset. The next steps were anchor identification with the “FindIntegrationAnchors” function, and data integration was performed with the “IntegrateData” function with the output of a merged assay (integrated) containing batch-corrected expression matrices. This was used in the downstream analysis. For the functions “RunUMAP”, “FindNeighbors”, and “FindClusters”, the exact numbers of PCs were calculated.

To quantify compositional differences between healthy and DMD organoids, cell type proportions were calculated for each cluster across all datasets. The statistical significance of proportion shifts was assessed using the chi-squared test. Effect sizes were quantified using Cramér’s V. To address technical heterogeneity across datasets (median genes per cell ranging from 371 to 2585), quality scores were assigned based on sequencing depth and incorporated into weighted effect size calculations. Confidence levels were stratified as high (quality score ≥ 4), moderate (3–4), or low (<3) based on combined technical quality assessments. Multiple testing correction was applied using Benjamini–Hochberg adjustment where appropriate. Statistical analyses were performed in R using the rstatix package, with the results documented in Appendix A.

Myogenic identity validation was performed using feature plots of canonical muscle markers generated in Seurat (v4.3). Core myogenic regulatory factors (PAX7, MYOD1, MYOG) were used to confirm myogenic progenitor cell identity and differentiation status, while structural markers (DES) and developmental myosin isoforms (MYH3, MYH8) were used to validate the fetal-to-neonatal maturation trajectory characteristics of the 12-week-old organoids. Feature plots were generated using the FeaturePlot function with UMAP reduction and split by dataset to demonstrate consistency across experimental conditions, with the results documented in Appendix A.

For the myogenic comparison of WT1 and DMD1, the integrated clusters for “PAX7” were selected and reprocessed to subdivide these populations into more detailed myogenic populations.

For the differential expression analysis between WT1 and DMD1 PAX7+ progenitors, Wilcoxon rank-sum tests with Benjamini–Hochberg correction were used. Stringent thresholds were applied: absolute log2FC ≥ 0.75, adjusted *p*-value < 0.01, min.pct ≥ 0.3 (minimum 30% of cells expressing the gene in either group), and min.diff.pct ≥ 0.25 (minimum 25% difference in expression frequency between groups).

#### 2.5.5. Developmental Score Analysis

The developmental score was calculated as described in [52]. The “AddModuleScore” function was used to calculate the embryonic and adult score, using a list of differentially expressed genes (DEGs) between adult and embryonic myogenic progenitor clusters. DEGs were selected from the supplementary information in [52], Table mmc3. The developmental score was further calculated by subtracting the embryonic score from the adult score. The embryonic and fetal datasets were filtered by applying a low threshold = 500, and the adult datasets were filtered by using a low threshold = 200–250 for genes per cell. In addition, the data was scaled with the vars.to.regress argument command, which included the “S.Score”, “G2M.Score”, “Stress” [50], and “total Count” to regress out the effects of the cell cycle, dissociation-related stress, and cell size/sequencing depth on all datasets. The myogenic subpopulation of organoids (WT1 + 2 and DMD1 + 2 + 3) and adult satellite cell datasets from [51] were selected by positive PAX7 expression. Embryonic and fetal (weeks 5 to 18) and adult satellite cell (years 7, 11, 34 and 42) scRNA-seq data were obtained from [52] (GSE147457), and adult satellite cell (year 25) scRNA-seq data were obtained from [51] (GSE130646), and adult satellite cell (year 20) scRNA-seq data were obtained from [53].

For statistical analysis, we compared developmental scores between groups using pairwise Wilcoxon rank-sum tests with continuity correction. Confidence intervals and effect sizes (Hodges–Lehmann estimator) were computed for each comparison. To correct for multiple testing, Benjamini–Hochberg adjustment was applied to all pairwise *p* values.

## 3. Results

Recent studies have shown that early myogenesis is already affected in DMD, wherein, at the somite stage, dysregulation of cell lineage markers and mitochondrial genes can be observed in in vitro differentiations [59]. Since skeletal muscle organoid systems are the only in vitro models that can establish a constant pool of resting myogenic progenitors, there has been a gap in research on healthy versus DMD-affected fetal myogenic progenitors thus far. To address this issue, scRNAseq analysis of 12-week-old skeletal muscle organoids (DMD1–3 and a healthy control) was performed.

We acquired three DMD iPSC lines, two from Children’s Hospital Boston [43] and one from UCSD [44], which had been fully characterized in the respective publications in which they were presented. Concerning the DMD geno- and phenotypes, DMD-iPS1 and DMD-iPS2 (CHB, Boston, MA, USA) were induced in a 6-year-old male patient with an identified deletion of exon45-52 of the dystrophin gene and a clinical DMD phenotype, while DMD3 (UCSD061i-65-1) (UCSD, San Diego, CA, USA) was induced in a 23-year-old male patient with an undisclosed dystrophin mutation, clinical DMD, and a dilated cardiomyopathy phenotype. We did not attempt to create an isogenic line from DMD1 as a control and instead used iPSCs derived from cord blood [45] as the control group, clearly constituting a limitation of our study.

The cells were differentiated according to the protocols we reported in the study by Mavrommatis et al. (2023) and specified in the work of Kindler et al. (2024) [38,46]. Microscopy images of organoid morphology at different time points of differentiation demonstrated a similar growth of DMD organoids as WT organoids (Appendix A). The three distinct stages of SMO development are as follows: (i.) predominant PAX3(+) precursors on day 17, (ii.) Pax7(+) cells towards the outside of the organoid from day 35 on, (iii.) MyoD(+) committed progenitors towards the outside as well as TITIN(+) skeletal muscle cells from week 8/9 on, which could be similarly detected by immunohistochemistry for DMD1 organoids as described by us previously [38] (Appendix A). These initial results encouraged us to avoid a detailed immunohistochemical analysis of DMD organoids, and we focused on deciphering their cellular composition by scRNA-seq analysis.

After the cells from the organoids were dissociated, the library was prepared using the 10X Chromium Kit 3.1 (10x Genomics B.V., Leiden, The Netherlands), followed by sequencing. The raw data were processed using Cell Ranger to annotate the single transcripts toward the human genome. The numbers of cells sequenced are listed in Appendix A. The median expressed genes per cell varied between 400 and 600. The literature recommends a minimum of 500 genes per cell [52]. Our estimated numbers are slightly lower. Since the mean reads per cell and the total number of reads had good rates, either the lower quality of a sample or technical issues during library preparation could have reduced the median number of genes per cell.

### 3.1. Quality Control for Single-Cell Data

Quality control is necessary to ensure that the data and results are of high quality and reliable for further downstream analysis. To exclude cells containing mitochondrial transcripts, the cutoff value for these cells has to be set at the beginning. After taking a closer look at the mitochondrial percentage, we set the limit for the mitochondrial cutoff to 10% (Appendix A). The cutoffs of the features were set as described in the Materials and Methods Section by looking at the MAD (median absolute deviation). The distribution of features in each cell in the datasets gives a general impression of the diversity within the average cells (Appendix A). Cells that contain too many features compared to their median value are not representative of the cells within a sample.

Cells with fewer genes/features (250 features per cell) reduce diversity within a population/sample. Previously published muscle dataset analyses have a wide range of feature cutoffs, ranging from 200 to 1000 genes per cell, as a minimum criterion [51,52]. To normalize the datasets, the “margin = 2” argument was included, which allowed normalization within the cell and not globally. Here, the result of normalization was more apparent due to the additional parameters.

The metrics are displayed in Appendix A. Compared with healthy controls, DMD SMOs showed lower average gene expression per cell. Subsequently, quality control was conducted to establish the best filter parameter, i.e., the one that reduced noise within each cell but retained important factors, which was applied using the Seurat package (v4.3). The typically used parameters are the minimum and maximum features per cell as well as the maximum percentage of mitochondrial genes. To calculate maximal features, we used a lenient cutoff of 5× median absolute deviations (MADs) of features [54] as well as a maximum of 10% mitochondrial RNA. As a general indicator, the minimum number of genes per cell should range up to 1000, a value that could not be reached with these samples. However, the trend of a decreasing number of genes in each cell (DMD1: 433, DMD2: 419, and DMD3: 371) compared to the WT counterparts (WT1: 2585 and WT2: 618) might be explained by the DMD phenotype. This factor could be problematic in some downstream analyses. As the minimum number of genes per cell was set to 250, all (except WT1 and WT2, for which the minimum number of genes was 500) the cells with a lower number of detected genes were excluded, reducing the number of analyzable cells.

### 3.2. Integrative Comparison of the 3D Skeletal Muscle Organoid Systems

To estimate the reproducibility of the 3D organoid systems, we integrated all five datasets. This integration minimized the batch and technically related differences during clustering. Integrated mapping of clusters confirmed that all the relevant clusters, which we previously reported, were present in the WT and DMD SMOs (Figure 1).

The manual annotation of the clusters based on the expression of key markers among them helped to identify cell populations (Figure 2A). PAX7 and MYF5 distinguished the myogenic progenitors, which could be divided into activated progenitors (expressing PAX7, MYF5, and CD44), mitotic myogenic progenitors (expressing PAX7, MYF5, TOP2A, and MKI67), and resting myogenic progenitors (expressing PAX7 and MYF5). Myocytes are cells with a strong expression of MYOG. The myotubes can be divided into two groups. One has a strong expression of MYH8, a marker for more mature myotubes. The absence of MYH8 with simultaneous co-expression of NEB and MYH3 indicates fewer mature myotubes, which are classified as embryonic (early) myotubes. Our focus was more on myogenic differentiation; the two clusters of neural progenitors were not further specified. Both clusters showed CD44 and SOX2 expression (Figure 2A).

Comprehensive transcriptomic validation of myogenic identity was achieved through expression analysis of canonical muscle markers across all organoid conditions (Appendix A). Developmental myosin heavy chain isoforms MYH3 (embryonic) and MYH8 (neonatal) validated the fetal-to-neonatal maturation trajectory achieved in the 12-week-old organoids, which is consistent with our developmental scoring analysis mapping organoids to fetal weeks 14–17 [38]. The structural marker DES (desmin) confirmed muscle fiber integrity across myotube populations. In addition, PAX7, MYOD1, and MYOG distributions confirmed the myogenic progenitor identity within the three clusters of resting, activated, and mitotic myogenic progenitors (Figure 1, Appendix A).

Evidently, there was a different cluster distribution among the WT and DMD datasets. The clusters for myotubes and the cluster for myogenic progenitors showed altered density (Figure 1), as analyzed and mapped in detail (Figure 2B, Table 1). Statistical validation of these compositional shifts using chi-squared tests revealed highly significant alterations across all comparisons (*p* < 10–22), with effect sizes ranging from moderate to very strong (Cramér’s V = 0.29–0.56). The most pronounced dysregulation occurred in the WT1-vs.-DMD2 comparisons (Cramér’s V = 0.55), reflecting a 65% reduction in resting progenitors and a 13-fold increase in the number of embryonic myotubes observed in this line (Appendix A).Our data show that the myotubes exhibited similar expression (WT1 SMO: 12.7%, DMD1 SMO: 14.02%), while the embryonic (early) myotubes tended to exhibit dramatically increased expression in DMD in comparison with the WT (WT1 + 2 SMO: <5.65%, DMD 1–3 SMO: >16.70%). The results demonstrate greater generation of myotubes (Table 1).

While the number of myotubes increased in the DMD SMOs, the number of clusters for myogenic progenitors (resting, mitotic, and activated) was reduced in the DMD SMOs. Among the WT SMOs, the percentage of all myogenic progenitors was relatively high (WT1 SMO: 48.49%, WT2 SMO: 40.80%), and the myogenic progenitor population in DMD was greatly reduced (DMD1 SMO: 24.91%, DMD2 SMO: 9.63%, DMD3 SMO: 12.88%) (Table 1). Playing an important role during muscle regeneration, fibro-adipogenic progenitors (FAPs) have been shown to be affected in dystrophic mice [60]. In the integrative analysis of all the SMO samples, no trend toward diseased SMOs was detectable (Table 1).

The FAP populations of the DMD1 and WT2 organoids were subdivided into four clusters. While clusters 0, 1, and 3 were more abundant in the DMD1 SMOs, cluster 2 was more prominent in the WT2 SMOs. Gene expression within each cluster varied across the cell lines. The expression of POLD1, a marker for collagen synthesis, was found in both groups of clusters 0 and 3, but only DMD FAP expressed it in cluster 2 (Appendix A).

Recent studies have described the expression of COL5A3, FST, MMP2, DCN, TIMP1, and LOX as being representative of proinflammatory FAPs in the absence of MMP14 [61]. While MMP14 was only absent in cluster 2 and in DMD cluster 0, the marker combination (COL5A3, FST, MMP2, DCN, TIMP1, and LOX) was found to be more highly expressed in cluster 0 of the DMD1 FAPs (Appendix A).

The expression of IGF1, a proinflammatory factor, which was only prominent in DMD1 FAPs, is in line with this finding (Appendix A).

### 3.3. DMD Myogenic Progenitors Exhibit an Altered Transcriptional Profile

Duchenne muscular dystrophy patients suffer from constant muscle degeneration, followed by increased fibrosis and impaired regeneration of SCs [18]. To investigate the alteration in myogenic progenitors in DMD organoids compared with their WT counterparts, the WT1 SMOs and DMD1 SMOs were integrated, and then subsets of the PAX7-expressing clusters were established. We focused our analysis on the DMD1 SMOs since they had been derived from the iPSC line with the defined dystrophin mutation, and their scRNA-seq dataset presented the best raw metrics among the three DMD datasets.

The subset was analyzed again, revealing six clusters (Figure 3A), representing the three major SC clusters (resting, activated, and mitotic). Differential expression analysis of PAX7+ progenitors identified 1338 genes passing stringent thresholds (|log2FC| > 0.75, adj. *p* < 0.01), including fibrosis-associated collagens (COL1A2/COL3A1, log2FC +1.9 to +2.2), dystrophin-interacting polarity markers (PARD3, log2FC −3.1), and stress–response transcription factors (JUN/FOS, log2FC +1.8 to +2.0) (Figure 3A, Appendix A). The percentage distribution of clusters in each dataset, comparing DMD1 MPs (myogenic progenitors) with WT1 MPs, showed an increase in cluster 0 (DMD1: 28.89%, WT1: 23.26%) and a dramatic increase in cluster 5 (DMD1: 28.89%, WT1: 4.61%) for the DMD1 MPs. While cluster 4 was nearly equally distributed in both the MPs (WT and DMD), clusters 1 (DMD1: 9.63%, WT1: 21.33%), 2 (DMD1: 10.62%, WT1: 19.08%), and 3 (DMD1: 8.64%, WT1: 19.19%) were more prominent in the WT1 MPs. The WT SMOs had a greater number of MPs than DMD1, but the committed cluster 5 exhibited a strikingly stronger contribution from DMD1.

To confirm the distribution of markers in the MP populations, Feature Plots were used, which clearly demonstrated PAX7 and MYH3 expression in separate clusters. These plots also indicated that the shape of the DMD1 MPs shifted more toward the mitotic and committed markers, while the PAX7 population was more prominent in the WT1 MPs (Figure 3B).

As a marker of asymmetric division, PARD3 was exclusively detected in WT dormant (cluster 0) and activated (cluster 2) MPs, pointing to a defect of asymmetric division in DMD (Figure 3C). In addition, polarizing factor MARK2 was not expressed. AURKA, which is necessary for the initiation and progression of mitosis, was detected in cluster 4 in the DMD and WT MPs; NOTCH3, an indicator of asymmetric cell division as well as a marker of the self-renewal of myogenic progenitors, was detected in dormant (cluster 0, 1), activated (cluster 2), and mitotic (cluster 4) MPs of both the WT and DMD MPs. JUN and FOS were observed in activated satellite cells. JUN and FOS were constantly expressed in the DMD MPs (Figure 3C).

The WT SMOs were clustered, and the PAX7 population was isolated using the subset function to apply it to the MMS to map its progenitor maturation. The outcome revealed a maturation grade overpassing fetal weeks 12 to 14 and reaching fetal weeks 17 to 18 (Figure 4) [38].

Furthermore, the MMS was checked for the WT1, WT2, DMD1, DMD2, and DMD3 datasets. While the WT1 and WT2 myogenic progenitors were clustered between fetal weeks 12–14 and 17–18, the DMD1 and DMD2 myogenic progenitor group could be found between fetal week 9 and fetal weeks 12–14 (Figure 4). The DMD3 myogenic progenitors tended to be as mature as the WT myogenic progenitors (Figure 4). Developmental delays in DMD progenitors were further quantified via Wilcoxon tests, showing highly significant differences (DMD1: *p* < 0.001, 95% CI [−0.069, −0.053]; DMD2: *p* < 0.001, 95% CI [−0.079, −0.053]) compared with the WT1, while DMD3 showed no significant differences (*p* = 0.60) (Appendix A).

## 4. Discussion

Duchenne muscular dystrophy is one of the most common genetic muscular disorders, affecting 1 out of 3500 newborn boys. Currently, there is no cure for this genetic disorder, and there are only limited therapeutic approaches that can extend patients’ lifespans and prevent death between the ages of 20 and 30. Therefore, disease modeling and ongoing research are essential for these patients.

The progressive loss of muscle over time makes patient-derived muscle biopsies a bottleneck for research over the course of the disease. Established techniques such as reprogramming cells into human-induced pluripotent stem cells greatly contribute to overcoming this bottleneck and are important ways to study the disease’s early phase, which often has not been studied precisely to date, as the disease is diagnosed at a later stage of development. Muscular dystrophies such as DMD are typically detected between the ages of 2 and 5. Here, the differentiation of hiPSCs into skeletal muscles offers insight into early myogenesis, as it has been shown to be affected in DMD [59].

### 4.1. Integrative Analysis Implementing the Robust Reproducibility of the SMO Model in Healthy and Diseased Cell Lines

Three-dimensional culture systems are used to improve the in vitro differentiation process, build a potential niche for progenitors, and improve the heterogeneity of the cells represented in culture systems [36,37,38,40,61,62]. However, insights into fetal myogenesis based on 2D in vitro approaches are very limited. Our SMO system not only recapitulates early myogenesis but also mimics fetal development, e.g., the PAX3 to PAX7 myogenic progenitor transition [63,64]. In addition, it establishes a population of resting myogenic progenitors, which indicates a stem cell niche [37,38]. The robust reproduction of the SMO system is an essential step in disease modeling. To examine the organoid differentiation of four additional iPSC lines, scRNAseq datasets were generated, and all five 12-week-old datasets showed a steady reproduction of all the relevant cell types represented in each organoid (Figure 1, [38]).

Interestingly, three cell groups stood out: resting myogenic progenitors, early myotubes, and myoblasts (Figure 2, Table 1). An organ-on-a-chip study indicated that more immature myotubes were formed from DMD myoblasts [65]. This is in line with our finding that the clusters expressed MYH3 as an early myotube marker since it was greatly increased in all three DMD lines compared with their WT counterparts.

Muscle regeneration also requires a pool of self-renewing MPs that either divide symmetrically or asymmetrically. In DMD, dystrophin, which is an essential player in asymmetric cell division, is missing or altered in conformation, resulting in a disturbed establishment of cell polarity in addition to mitotic stress. This can lead to a loss of progenitor cells over time since these cells exit the cell cycle during regeneration [18,21,66]. In addition, constant degeneration of muscles in DMD triggers persistent satellite cells [67]. The impaired maintenance of myogenic progenitor cells combined with an increased need for regeneration leads to a reduced pool of resting myogenic progenitor cells in DMD [63,68]. The scRNAseq data for the healthy and DMD SMOs confirmed this hypothesis, with an increased resting myogenic progenitor pool in the WT and a trend toward an increased myoblast pool in DMD (Figure 2, Table 1).

### 4.2. Fibro-Adipogenic Progenitors

Muscle mass maintenance is achieved via the interplay of various types of cells, like immune cells, fibro-adipogenic progenitors (FAPs), and others [69]. FAPs are a muscle-resident stem cell population interacting with the cells in the SC niche and responding to changes in their environment to assist in muscle repair. During this process, these cells can differentiate into adipocytes or fibroblasts. FAPs have recently become an increasing focus of research because they play an important role in muscle homeostasis and regeneration under healthy and pathological conditions; their changed behavior in muscular dystrophies like DMD is of particular interest [70,71]. In this study, there was no difference in the number of FAPs in the healthy and DMD SMOs, as their percentages varied between the healthy and DMD groups (Table 1).

But the FAP populations within DMD tended to express more collagens, encoding genes responsible for collagen synthesis, an indicator of increased fibrosis, while the WT FAP populations clearly lacked these genes, indicating that they were in the resting phase (Appendix A). In combination with genes responsible for ECM production, like MMP2 and TIMP1, degradation is an indicator of activated FAPs. The constant activation of FAPs, which are seen in DMD SMOs, is typical for chronic injuries and muscle degeneration [69].

Recent analyses of FAPs in mice identified proinflammatory subpopulations specific to DMD, a finding that aligns with the elevated COL5A3/MMP2 expression observed in the DMD FAPs (Appendix A). While chronic inflammation is a known driver of fibrotic remodeling in DMD [9], the direct link between FAP activation and progenitor depletion in human organoids remains speculative and requires functional validation [72] Our previous comprehensive cell–cell communication analysis of healthy skeletal muscle organoids revealed that myogenic progenitors receive prominent ECM-related signals from fibro-adipogenic progenitors, while FAPs respond to signals from activated myogenic progenitors [38]. In our DMD organoid analysis, we observed both elevated FAP activation (increased levels of proinflammatory markers: IGF1, COL5A3, and MMP2) and concurrent myogenic progenitor dysregulation (reduced levels of resting progenitors: 34.49% to 7.65–19.65%). This evidence indicates a disrupted FAP–myogenic communication axis in DMD, where hyperactivated FAPs may contribute to the observed developmental arrest through altered ECM signaling. The upregulation of fibrotic markers (COL1A2, COL3A1, and TIMP1) in DMD myogenic progenitors further supports this dysregulated crosstalk, potentially perpetuating the cycle of impaired regeneration characteristic of DMD pathology.

### 4.3. Myogenic Progenitors

There was a lack of in vitro approaches that established a stable platform for myogenic progenitor maturation until recently, when skeletal muscle organoid systems were established [37,38,40,41]. The heterogeneity of cell types within the SC niche has been shown to be essential for the proper regeneration and development of muscles and can be mimicked manually via a combination of different progenitor cells. The scRNAseq results showed that besides the neural progenitors and FAPs, specifically myogenic progenitors in their different dormant, activated, and mitotic states could be identified [38].

This study showed that the subclusters between the DMD and WT SMOs were clearly different, with increased percentages of committed MPs in the DMD SMOs (Figure 1 and Figure 2, Table 1). The satellite cells in DMD were affected by constant activation, which can be explained by the increased quantity of committed progenitors in DMD [16]. This hypothesis is supported by the expression of FOS and JUN in all the clusters of DMD myogenic progenitors as markers of activated progenitors (Figure 3C).

Interestingly, only two clusters of WT MPs expressed PARD3 (Figure 3C). Asymmetric cell division is thought to be impaired in DMD satellite cells, as they fail to properly establish cell polarity, which is dependent on dystrophin expression [18]. The DMD1 sample did not show any PARD3 expression, which is a marker indicating a defect in asymmetric division. DMD defects in asymmetric satellite cell division have been predominantly evaluated in the murine system. In light of our scRNA-seq-based observations, it will be interesting to explore PARD3 alterations in human DMD further.

### 4.4. Myogenic Maturation Score

Myogenesis spans from early embryonic development to postnatal maturation. Until recently, little was known about myogenic development in humans. Recent scRNAseq studies covering different time points of muscle development are beginning to clarify these ranges [52]. Samples of comparable age are especially helpful for encoding the heterogeneity of myogenic progenitors and satellite cells [51,53,73]. Using embryonic expression and adult satellite profiles, the beginning and endpoint of maturation during development can be marked. The distinct genetic expression profile of each sample can be used to build a roadmap of myogenesis [52]. Through these means, the efficiency of in vitro protocols for differentiating mature myogenic progenitors can be evaluated. The organoid system could derive myogenic progenitors with a maturation grade similar to that of late fetal MPs, namely, weeks 14 and 17 (Figure 4) [38]. In contrast to the WT MPs, the dystrophic progenitors DMD1 and DMD2 showed significantly decreased maturation in terms of the myogenic maturation score, which was mapped at fetal week 9. The DMD3 sample also showed decreased maturation, albeit scoring closer to the WT MPs, showing different results in the developmental maturation analysis in comparison with DMD1 and DMD2 (Figure 4). It is worth evaluating the degree to which the different dystrophin mutations reported in the studies by Panopoulos et al. (DMD3) and Park et al. (DMD1, DMD2) contribute differently to progenitor maturation.

Currently, we can only speculate that DMD progenitors with different origins also show heterogeneity, as observed for adult satellite cells via scRNA-seq analyses [53]. In this context, we must acknowledge the limitations of our overall study design concerning the statistical significance of all the conclusions stated above. We only presented data from one technical replicate for each of the three DMD lines versus two technical replicates for the non-isogenic control group. Further provision of organoid scRNA-seq datasets derived from DMD patients and other muscle dystrophy patients by us and other groups will validate whether our observations can be underpinned. Our study design could be further improved via the derivation and use of isogenic iPSC lines derived from DMD patients as controls for organoid differentiation, as we have outlined previously [74].

The absence of isogenic controls increases the chances that genetic background variations besides the DMD1 iPSC-documented exon 45-52 deletion influence our major scRNA-seq-data-based claim of a developmental delay in myogenic progenitor maturation. (i.) The absence of a functional dystrophin protein in DMD patients and the corresponding mdx mice was first described in 1987 by Hoffman, Brown, and Kunkel [75]. The human DMD gene is one of the largest genes, with 2.4 Mbp encompassing approximately 79 exons, for which over 7000 mutations have been identified (Leiden Muscular Dystrophy pages; https://www.dmd.nl/ (accessed on 20 June 2025)). Due to its size, the probability of dystrophin mutation is likely higher than for any other gene that could also have affected our non-isogenic control iPSCs. (ii.) Dumont et al. reported that polarized dystrophin expression interacts with the cell polarity-regulating kinase MARK2 in murine satellite cells and abrogates asymmetric satellite cell division, resulting in the loss of myogenic progenitors [18]. Dystrophin ablation does not have to be the only cause of their observed satellite cell division dysfunction. Mutations in other genes, e.g., direct mutations in MARK2, might result in similar changes in myogenic progenitor pools. As a further example, Dystroglycan mutations can also cause DMD-like phenotypes [76]. In a non-isogenic control DMD disease modeling setting like ours, there is a higher chance that the observed claims are falsely attributed to the reported exon 45-52 deletion, while they may also originate from undetected mutations of other genes. Future studies with isogenic controls or larger cohorts stratified by mutation type would help to further distinguish disease-specific from background-related variations.

Compared with 2D protocols [24,26,27], the SMO system better supports the maturation of myogenic progenitors as cells patterned by these protocols, as they only yield an embryonic to fetal developmental score toward fetal weeks 8 and 9 [38,52]. These findings indicate that the 3D skeletal muscle organoid system evaluated and applied in this study provides a state-of-the-art model for investigating human myogenic development and myogenic progenitor maturation in healthy and diseased/dystrophic samples.

## 5. Conclusions

Our study investigated the changes in myogenesis in DMD-affected SMOs. The ability to establish a stem cell niche for myogenic progenitors makes this organoid system very valuable for culturing myogenic progenitors over long periods of time without sacrificing self-renewal capacity. SMOs mimic and propagate the different myogenic progenitor states (dormant, activated, and mitotic) as well as their maturation. In 2D protocols, the population of myogenic progenitors decreases over time, and therefore, the later stages of maturation of the muscle progenitor cells can only be studied to a limited extent. The constant loss of progenitor cells in 2D protocols also indicates a missing niche for uncommitted PAX7 cells. Therefore, 2D protocols are not as suitable for studying myogenic progenitor cells in muscle dystrophies such as DMD, particularly when the aim is to examine these cells over the course of their maturation process, which is crucial for understanding disease progression and developing targeted therapies.

Our data reveal a developmental delay in DMD organoids, with a significantly higher proportion of embryonic myotubes (Table 1) and reduced PAX7+ progenitor pools (WT1: 48.49% vs. DMD1: 24.91%). The strong effect sizes (with a Cramér’s V = 0.56) emphasize the robust DMD pathology in our datasets despite technical limitations. While FAP activation (COL5A3, MMP2) is correlated with this delay (Appendix A), causal links between FAP signaling and myogenic immaturity remain speculative and require functional validation via cytokine blocking or co-culture assays.

As reported, constant satellite cell activation and a lack of satellite cells lead to a reduced pool of progenitors, as observed in the DMD SMOs. All three progenitor groups decreased in number. A closer look at the genes in the DMD1 MPs confirmed that there was a greater pool of activated myogenic progenitors in the DMD SMOs in comparison to the WT. The constant activation of DMD myogenic progenitors raises the following question: to what degree, if at all, can uncommitted PAX7 progenitors be sustained, as demonstrated for WT Pax7 progenitors in vitro as well as in the pathophysiology of the disease?

Recent studies using the murine system have linked satellite cell quiescence and maintenance of the niche to physical compression [77,78,79]. Stretching and compressing DMD vs. WT organoids might release the more immature DMD myogenic progenitors that we reported herein. However, since our myogenic progenitor cells mainly grow on the outside of SMOs [38], we would currently not expect that the absence of dystrophin changes tissue tension towards the organoid outside considerably.

To conclude, myogenic progenitor maturation toward the satellite-cell state can be mapped to examine the maturation grade reached in healthy and DMD SMO-derived myogenic progenitors. Since the score measures the development from embryonic myogenic progenitors to satellite cells, the observed maturation delay of myogenic progenitors in DMD might also indicate developmental deceleration.

## Figures and Tables

**Figure 1 cells-14-01033-f001:**
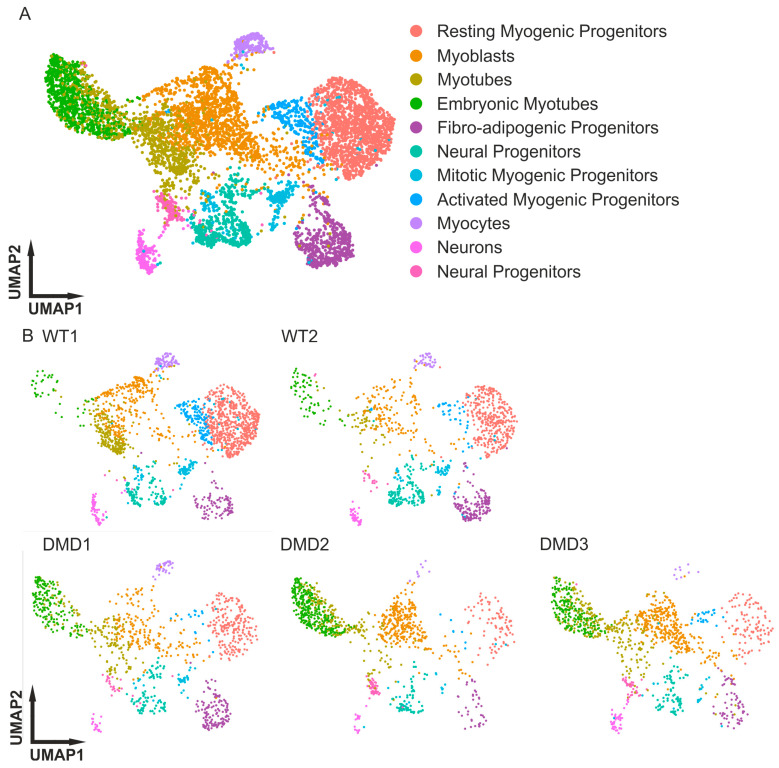
Integrated scRNA-seq analysis of healthy control and DMD SMOs. (**A**) UMAP cluster of integrated scRNA-seq analysis showing cell type distribution in the SMOs. (**B**) UMAP analysis splitted into individual datasets to show cluster distribution between healthy and DMD SMOs. The DMD SMOs comprise a reduced population of resting myogenic progenitors and a greater pool of early embryonic myotubes compared to the healthy controls (DMD1, DMD2: [43]; DMD3: [44]; WT: [45]). SMOs = skeletal muscle organoids.

**Figure 2 cells-14-01033-f002:**
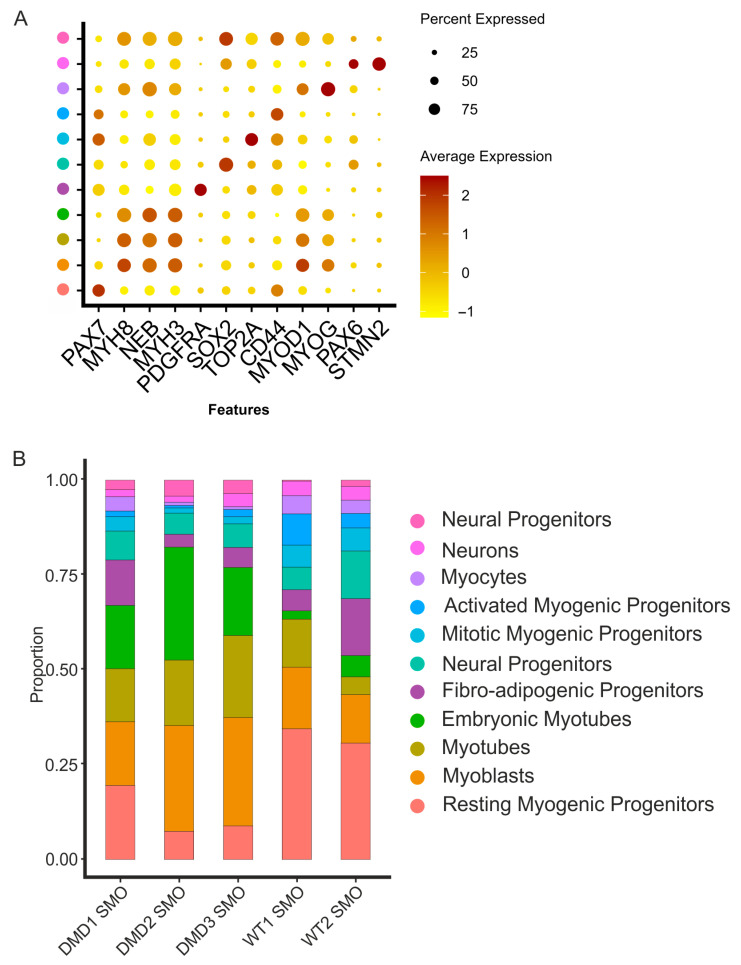
Identification of the types of cells within the organoid systems and altered distribution of clusters for diseased and healthy SMOs. (**A**) Dots showing marker genes for cluster identification. Clusters were identified via the upregulation of key markers (PAX7, MYH8, NEB, MYH3, PDGFRA, SOX2, TOP2A, CD44, MYOD1, MYOG, PAX6, and STMN2). (**B**) Distribution of clusters within the WT vs. DMD SMOs. The percent distribution of cells across the different datasets revealed increased clusters of myogenic progenitors in the WT SMOs. The number of myoblast clusters tends to be higher in DMD SMOs, and the number of embryonic (early) myotubes increases in all DMD SMOs.

**Figure 3 cells-14-01033-f003:**
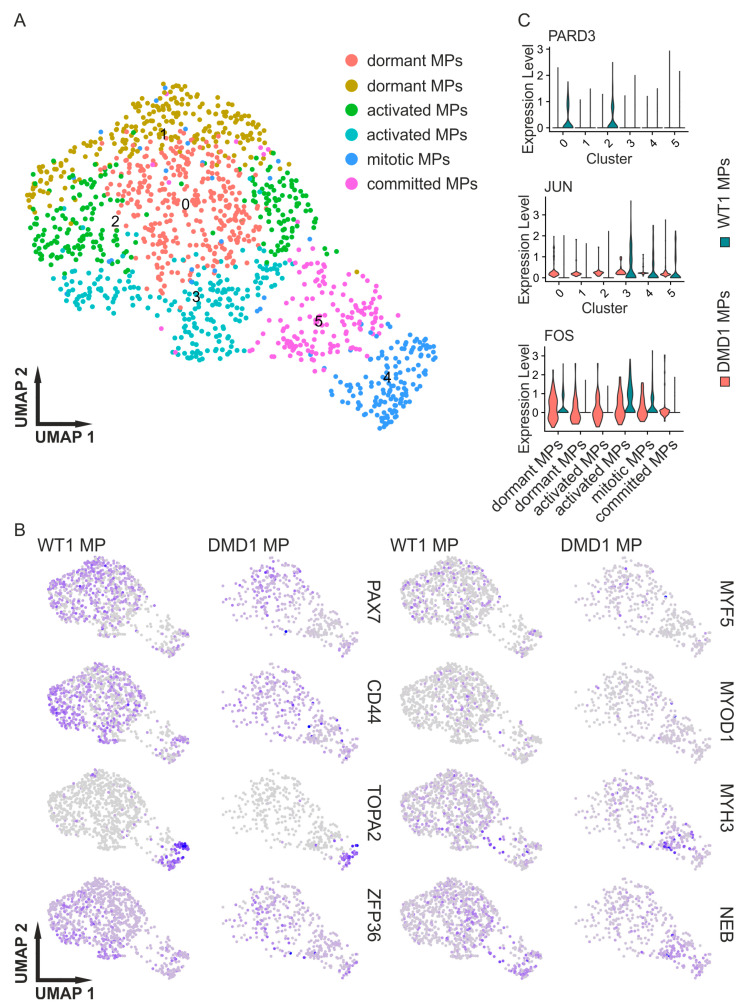
Investigation of DMD-affected myogenic progenitors. (**A**) Results of an integrated scRNA analysis of organoid-derived myogenic progenitors of WT and DMD iPSCs showing up to 6 subclusters. (**B**) Feature plots of representative genes of dormant, activated, and committed myogenic progenitors (PAX7, CD44, TOP2A, ZFP36, MYF5, MYOD1, MYH3, and NEB). Clusters that expressed PAX7, MYF5, and CD44 were more prominent in the WT, while clusters expressing NEB and MYH3 were more prominent in the DMD MPs. (**C**) Example expression of the WT and DMD clusters of PARD3 (an asymmetric cell division marker), JUN, and FOS as markers for activated MPs (cluster numbers corresponding to subfigure A). MPs = myogenic progenitors.

**Figure 4 cells-14-01033-f004:**
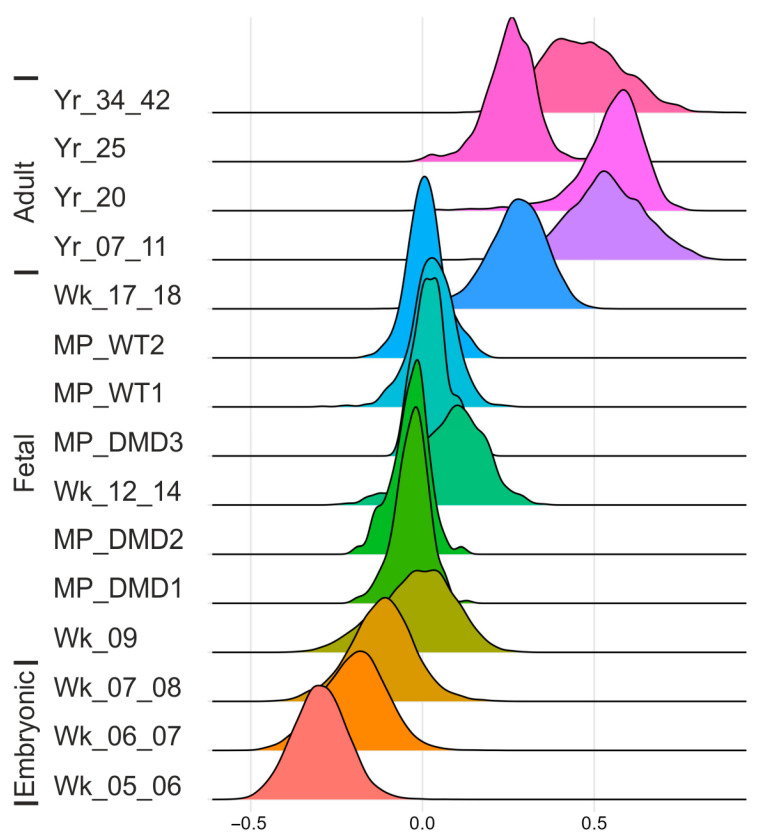
Developmental scores of skeletal muscle organoid myogenic progenitors. The distribution of myogenic progenitors in the maturation score indicates delayed development of DMD1 and DMD2 MPs compared with WT1 and WT2 MPs across in vivo stages. DMD3 shows the same maturation grade as the WT MPs. The score is based on the difference between upregulated satellite cells and embryonic markers from human reference atlases for weeks (Wk) 5 to 18 in the embryonic and fetal stages and years (Yr) 7 to 42 for adult satellite cells and SMO-derived myogenic progenitors. The colours in the ridge plot were generated using the default settings of the ridgeplot function and are representing the different developmental maturation stages. MPs = myogenic progenitors.

**Table 1 cells-14-01033-t001:** Percentage distributions of cell types within the SMOs for various healthy and diseased cell lines.

Sample	WT1 SMO	WT2 SMO	DMD1 SMO	DMD2 SMO	DMD3 SMO
Cell Type					
**Neural Progenitors**	0.22%	1.63%	2.49%	4.23%	3.41%
**Neurons**	3.62%	3.51%	1.75%	1.44%	3.32%
**Myocytes**	4.89%	3.51%	3.78%	0.90%	0.77%
**Activated Myogenic Progenitors**	8.18%	3.76%	1.48%	0.63%	1.88%
**Mitotic Myogenic Progenitors**	5.82%	6.16%	3.78%	1.35%	1.96%
**Neural Progenitors**	5.88%	12.57%	7.56%	5.58%	6.14%
**Fibro-Adipogenic Progenitors**	5.60%	14.97%	11.90%	3.42%	5.20%
**Embryonic Myotubes**	2.20%	5.65%	16.70%	29.79%	17.99%
**Myotubes**	12.74%	4.62%	14.02%	17.19%	21.74%
**Myoblasts**	16.36%	12.75%	16.79%	27.81%	28.56%
**Resting Myogenic Progenitors**	34.49%	30.88%	19.65%	7.65%	9.04%

## Data Availability

The original data presented in this study are openly available in the NCBI GEO database at https://www.ncbi.nlm.nih.gov/geo/query/acc.cgi?acc=GSE277637 (accessed on 1 July 2025) under the reference number GSE277637 (DMD1, DMD2, DMD3, WT2) and at https://www.ncbi.nlm.nih.gov/geo/query/acc.cgi?acc=GSE147514 (accessed on 1 July 2025) under the reference number GSE147514 (WT1).

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
