# Peer review of "Duchenne Muscular Dystrophy Patient iPSCs—Derived Skeletal Muscle Organoids Exhibit a Developmental Delay in Myogenic Progenitor Maturation"

_cells, 2025, doi:10.3390/cells14131033_

Round 1
Reviewer 1 Report
Comments and Suggestions for Authors
The study enhances the existing body of knowledge in the field of SMO for DMD disease modeling overcoming the limitations of human samples and for testing therapies for DMD potentially targeting progenitor cell maintenance, maturation, and fibrosis reduction. The article is technically sound and well written. Anyhow, its limited to a small sample size, and needs further validation to strengthen the conclusions and broaden the applicability of the findings.
Strenghtness
The SMO successfully replicate early myogenesis and stem cell niche, providing a valuable tool for studying DMD and other muscular dystrophies.
The study introduces a myogenic maturation score to map the maturation grade of progenitors, enabling comparisons between healthy and DMD samples across developmental stages.
Limitations
The study uses one technical replicate for each of the three DMD iPSC lines and two replicates for the healthy control group. This small sample size limits the statistical significance of the findings.
Heterogeneity of SMO from either WT and DMD samples is observed, which in DMD may be influenced by variations in dystrophin mutations. Please provide some details (not only the references) on the subjects from which the iPS were derived: i.e. age, sex, the type of mutation and the disease onset. Because of this heterogenicity, the selection of a certain SMO that has been made for some analyses (i.e the integration of the WT1 and DMD1 to investigate the alteration of myogenic progenitors) should be justified and the result and overall conclusions must be not generalized.
Minor points
In Results, details reported in lines 263-264 and 276-277 should be included in the Methods section.
Please change the green color for embryonic and fibro-adipogenic cell types, because they are too similar at the moment.
Figure 4 is not clear to me. What do represent the arrows? What the different colors represents?
There are many abbreviations and acronymous in the paper. Please provide an abbreviation list for those that are not introduced in the text.
Reviewer 2 Report
Comments and Suggestions for Authors
This manuscript presents a 3D skeletal muscle organoid (SMO) model derived from human iPSCs to investigate Duchenne Muscular Dystrophy (DMD)-associated myogenesis. The study provides valuable insights into early myogenic events, the behavior of myogenic progenitor cells and fibro-adipogenic progenitors (FAPs), and comparative developmental maturity between DMD and WT muscle lineages. The integration of single-cell RNA sequencing (scRNA-seq) provides high-resolution analysis of cell types and differentiation trajectories.
While the study addresses a clinically relevant problem using cutting-edge methodologies, substantial revisions are needed. Specifically, the manuscript would benefit from clearer articulation of novelty, more rigorous data interpretation, improved statistical handling, and better integration of limitations. In its current form, several claims are insufficiently supported by the data.
Major Comments
- Clarity on Novelty and Model Improvement
- The authors should explicitly articulate how their SMO model improves upon or differs from previously reported 3D muscle or DMD organoid systems. This clarification should be added to the abstract and introduction.
- Statistical and Experimental Rigor
- Replicates: The manuscript reports only one technical replicate per DMD iPSC line and two for WT, limiting statistical robustness. This limitation should be discussed clearly, and interpretations adjusted accordingly.
- Statistical Testing: Quantitative comparisons (e.g., maturation scores, progenitor proportions) must include appropriate statistical analyses with p-values and confidence intervals.
- Validation of Organoid Identity: The use of brightfield imaging is noted, but immunostaining for myogenic markers (e.g., MYOD, MYOG, Desmin, Dystrophin, MyHC) should be included or referenced to confirm cellular identity within SMOs.
- Cell Input Normalization: The scRNA-seq methods mention a broad range of input cells (6.6×10⁵ to 1×10⁶). Clarify whether cell input was normalized per line prior to library preparation to avoid biases.
- Essential Data in Main Figures: Several critical analyses are relegated to supplementary materials (e.g., FeaturePlots, developmental scores). These should be included in the main figures for clarity and transparency.
- Isogenic Controls
- The absence of isogenic controls is a major limitation. This issue is briefly acknowledged but should be addressed earlier in the results or discussion with a more thorough examination of how this may affect interpretations.
- Overinterpretation of Transcriptomic Data
- Claims such as “functional immaturity,” “defective asymmetric division,” and “developmental delay” are not fully supported by the transcriptomic data. These should be rephrased as hypotheses or possibilities unless supported by functional assays or validated protein-level evidence.
- Introduction Clarity (Line 256–261)
- The statement that “early myogenesis is already affected in DMD” is too vague. Specify which stages (e.g., paraxial mesoderm, somite, myotome) are implicated, and reference supporting literature to justify the current study.
- Data Quality Concerns (Line 297–302)
- A low number of detected genes per cell, especially in DMD3 (371 genes), raises concerns about sequencing depth and data quality. Discuss how this issue was accounted for in downstream analysis, especially clustering.
- scRNA-seq Analysis Transparency
- Provide more detail on:
- Clustering methodology and resolution settings.
- Statistical thresholds for differential expression (e.g., adjusted p-values, log2FC cutoffs).
- Handling of batch effects or donor variation.
- Validation of cell-type annotation via marker genes or protein-level confirmation.
- Provide more detail on:
Minor Comments
- Grammar and Readability
- The manuscript would benefit from professional editing. Examples from the abstract:
- “Affecting worldwide 1 in 3500-5000 new-born boys is causing constant skeletal muscle weakness and loss” → Suggested revision: “Duchenne muscular dystrophy (DMD), which affects 1 in 3,500 to 5,000 newborn boys worldwide, is characterized by progressive skeletal muscle weakness and degeneration.”
- “Dystrophin's mutation or absence... is discussed to disrupt SC asymmetric division” → Suggested revision: “The absence or mutation of dystrophin in DMD is hypothesized to impair satellite cell (SC) asymmetric division…”
- The manuscript would benefit from professional editing. Examples from the abstract:
- FAP-Myogenic Interaction
- While FAPs are mentioned, it is unclear whether observed ECM/fibrotic gene signatures in DMD SMOs correlate with altered myogenic progenitor behavior. This relationship should be discussed or tested more directly.
- Justification of iPSC Lines
- The rationale for selecting specific iPSC lines (mutation type, age, clinical severity) should be provided to improve reproducibility and biological relevance.
- Matrigel vs. Geltrex Use
- The manuscript mentions both coatings; clarify whether these were used interchangeably and how potential batch variability was managed.
- Differentiation Protocol Overview
- Although protocols are cited ([38], [43]), a brief summary of key differentiation steps should be included to improve self-containment.
- Terminology Consistency
- Cell types are inconsistently referred to as “SMPs,” “SCs,” and “myogenic progenitors.” Clarify definitions and use consistent terminology throughout.
- Discussion Structure
- Improve flow in the discussion by distinguishing clearly between data-supported conclusions and speculative ideas.
Comments on the Quality of English Language
Please see comments above.
Reviewer 3 Report
Comments and Suggestions for Authors
The study titled “Duchenne Muscular Dystrophy Patient iPSC-Derived 2 Skeletal Muscle Organoids Exhibit a Developmental 3 Delay of Myogenic Progenitor Maturation” by Holm’s team is interesting. The researchers employ organoid technology to show that satellite cells derived from dystrophic patients (DMD) produce different clusters of cells at a more immature developmental stage when compared to cells from healthy individuals. Furthermore, the study reveals that fibro-adipogenic progenitor cells (FAPs) are activated in DMD organoids, leading to the secretion of pro-inflammatory and pro-fibrotic extracellular matrix proteins, which alter the satellite cell niche.
However, there are several concerns with the manuscript in the current format. The material and methods section lacks sufficient detail to allow readers to replicate the findings. I recommend that the authors revise this section carefully to include the necessary information. Additionally, the Results section should be clearly presented based on statistical analysis. Furthermore, the text requires a thorough revision for reference formatting, syntax and spelling. Below, I have provided my comments on each section of the manuscript.
Major Comments
Material and methods:
- Integration was used in the results section to produce data described in items 3.2 and 3.3. Several paragraphs mention integration. However, it is unclear how integration was performed to produce individual data. Please clarify the methodology of integration used to produce the data. The Supplementary Material and methods section can be used for a detailed explanation of the integration methodology.
- In Figures 1 and 3A, satellite cells were grouped in clusters. Please clarify the methodological strategy of clustering and which markers were used.
- In Table 1, data are presented as a percentage of the distribution of cell types within healthy and dystrophic SMO. The narrative in the results section analyses the differences among the values in Table 1 (Paragraph is shown below). However, statistical outcomes were not provided for the comparisons made in the text. Please clarify how the data was compared and provide statistical outcomes.
Line 327:
“Our data show that the myotubes had a similar expression (WT1 SMO: 12.7%, DMD1 327 SMO: 14.02%), while the embryonic (early) myotubes tended to have a dramatically increased expression in DMD compared to WT (WT1+2 SMO: < 5.65 %, DMD 1-3 SMO: > 329 16.70%). The results demonstrate a larger generation of myotubes (Table 1).”
- How are the cutoffs determined. Which are the cutoffs? Line 280 “The minimum and maximum cutoffs of features are the next filter criteria”.
Attention to referencing format, English language, text format:
There is a generalised lack of attention to the text referencing format, English syntax and meaning.
- The manuscript shows a lack of attention regarding in-text referencing. There are formatting inconsistencies observed in the text. The format required by this journal is in-text numbers, but on several occasions, the manuscript refers to the author's name. To mention a few examples, in the material and methods paragraph listing the cell lines, the author's reference name is given; in line 445, the author's name (Rubenstein) is given instead of the reference number.
- In other lines of the text, references could have been added. To mention a few examples, lines 405-406, Line 276, 282 add reference. Please revise the text to adopt the correct referencing format.
- Revise English syntax; some phrases do not make sense. To mention a few examples, line 263, line 265. In the discussion section, line 526.
- Check the spelling (e.g. Figure S2 legend should say Violin Plots)
- The manuscript uses a large number of abbreviations. Revise the manuscript to make sure the full description is available the first time it appears in the text; no repetition of the abbreviation's meaning is necessary (line 308). All abbreviations should be given in the text, Line 121, SHH.
Results
- The result section contains extensive discussion content. Results should be rewritten to focus on presenting facts that will address the hypothesis.
Discussion
Muscle tissue development is dependent on perpendicular tension. The genetic deficiency of dystrophin plays a role in tissue tension and development. Do the authors believe that this may have contributed to the more immature developmental aspect of their model? If so, have they tried using a model that offers perpendicular focal points of tension? What limitations should be considered?
Comments on the Quality of English Languagethe text requires a thorough revision for reference formatting, syntax and spelling. Below, I have provided my comments on each section of the manuscript.
Round 2
Reviewer 2 Report
Comments and Suggestions for Authors
The authors have revised the manuscript in response to the initial set of comments, and I appreciate the efforts to address the concerns raised. Several of the major points have been reasonably clarified or acknowledged, particularly with respect to the study’s limitations regarding replication and the lack of isogenic controls. However, there remain some key areas that have not been sufficiently addressed or clarified in the revision. My detailed assessment is as follows:
- Replicates and Isogenic Controls
The authors have appropriately acknowledged the limited number of technical replicates (one per DMD iPSC line and two for WT) and the absence of isogenic controls. They have added discussion around these points in the revised version. While these limitations remain significant for interpreting the robustness of the findings, the acknowledgment and tempered conclusions make the current discussion acceptable. However, the original request for a more thorough examination of how the absence of isogenic controls may specifically influence interpretations has not been fully addressed. A deeper consideration of how genetic background variability may have affected observed outcomes would strengthen the Discussion section. - Cell Input Normalization for scRNA-seq
The authors did not respond to the core concern of this comment. The original question related to whether the variation in input cell numbers (ranging from 6.6×10⁵ to 1×10⁶) was normalized across lines prior to library preparation to reduce bias. Instead, the authors described their statistical analysis pipeline, which, while informative, does not clarify whether input normalization or control for technical variability was performed at the library preparation stage. This point remains unaddressed and requires clarification, as it could directly impact cell representation and downstream interpretation of the scRNA-seq results. - Validation of Organoid Identity
While the manuscript mentions the use of brightfield imaging, no immunostaining data or references to confirm myogenic identity of the SMOs (e.g., MYOD, MYOG, Desmin, Dystrophin, MyHC) have been provided. These markers are critical for verifying the differentiation status and cellular composition of the organoids. Without such data the characterization of SMOs remains incomplete. I recommend that the authors include representative immunostaining images to support the cellular identity claims.
Most of the prior comments have been addressed satisfactorily. However, the manuscript still requires clarification or additional information on three important aspects: (1) deeper discussion on the absence of isogenic controls, (2) explanation of input normalization for scRNA-seq, and (3) immunostaining-based validation of SMO identity. These points are critical to support the rigor and reproducibility of the study. I encourage the authors to address these remaining issues before the manuscript can be considered for acceptance.
Reviewer 3 Report
Comments and Suggestions for Authors
Dear Authors,
Thank you for your thorough and thoughtful revisions to the manuscript. I appreciate the care taken to address the feedback provided in the previous round. The changes have notably improved the clarity and quality of the work.
Author Response
Thanks again for your comments, which clearly helped to improve our manuscript.
Round 3
Reviewer 2 Report
Comments and Suggestions for Authors
The authors have satisfactorily addressed my comments.